# Randomised controlled trial on the effect of video-conference cognitive behavioural therapy for patients with schizophrenia: a study protocol

Masayuki Katsushima [ID] ,[1,2] Hideki Nakamura,[1,3] Hideki Hanaoka,[4,5] Yuki Shiko [ID] ,[4] Hideki Komatsu,[6] Eiji Shimizu [ID] [7,8]

For numbered affiliations see end of article.

**Correspondence to**
Masayuki Katsushima;
m.katsushima@thu.ac.jp

## ABSTRACT

**Introduction** Cognitive behavioural therapy for psychosis (CBTp) has demonstrated effectiveness in reducing positive symptoms, improving depression, enhancing coping skills and increasing awareness of illness. However, compared with cognitive behavioural therapy for depression and anxiety, the spread of CBTp in clinical practice is minimal. The present study designed a randomised controlled trial (RCT) research protocol to evaluate whether real-time remote video-conference CBTp (vCBTp) could facilitate access to psychosocial interventions and effectively improve symptoms compared with usual care (UC) for patients with schizophrenia.

**Methods and analysis** This exploratory RCT will consist of two parallel groups (vCBTp+UC and UC alone) of 12 participants (n=24) diagnosed with schizophrenia, schizoaffective disorder or paranoid disorder, who remain symptomatic following pharmacotherapy. Seven 50-min weekly vCBTp interventions will be administered to test efficacy. The primary outcome will be the positive and negative syndrome scale score at week 8. The secondary outcome will be the Beck Cognitive Insight Scale to assess insight, the Patient Health Questionnaire-9 to assess depression, the Generalised Anxiety Disorder-7 to assess anxiety, the 5-level EuroQol 5-dimensional questionnaire to assess quality of life and the Impact of Event Scale-Revised to assess subjective distress about a specific stressful life event. We will take all measurements at 0 weeks (baseline) and at 8 weeks (post-intervention), and apply intention-to-treat analysis.

**Ethics and dissemination** We will conduct this study in the outpatient department of Cognitive Behavioral Therapy Center at Chiba University Hospital. Further, all participants will be informed of the study and will be asked to sign consent forms. We will report according to the Consolidated Standards of Reporting Trials.

**Trial registration number** UMIN000043396.

## INTRODUCTION

Based on the fifth edition of the Diagnostic and Statistical Manual of Mental Disorders (DSM-5), schizophrenia is a mental disorder associated with functional disability, poor quality of life and early death.[1] Schizophrenia is a severe, lifelong mental disorder that affects approximately 1% of the world's population.[2] The illness tends to develop between the ages of 16 and 30 years.[3] Individuals with schizophrenia use a substantial amount of healthcare services. This condition imposes a significant economic burden on both the patients and their families, and on society as a whole.[4]

### Standard treatment for schizophrenia

The treatment of schizophrenia is multidimensional and comprehensive. The first treatment option is pharmacotherapy with antipsychotic medications and psychological interventions such as cognitive-behavioural therapy, as recommended by the National Institute of Health and Clinical Excellence.[5] Antipsychotic medications are commonly used for patients with schizophrenia primarily to reduce positive symptoms such as hallucinations and delusions by regulating the activity of dopamine and other nerves in the brain. In recent years, atypical antipsychotic

prescriptions have been promoted. They effectively improve positive and negative symptoms and cognitive function but may cause side effects in some patients.

### STRENGTHS AND LIMITATIONS OF THIS STUDY

⇒ This study will be a pilot randomised controlled trial to test the effectiveness of real-time remote cognitive behavioural therapy using a video-conference system for outpatients diagnosed with schizophrenia.
⇒ The intervention comprises seven sessions, with elements of psychoeducation, case formulation and cognitive restructuring.
⇒ Remote cognitive behavioural therapy in this study has the potential to reduce the burden of access for receiving cognitive behavioural therapy for schizophrenia.
⇒ This is a single-centre pilot study and statistical analysis between the usual care (UC) and video-conference cognitive behavioural therapy for psychosis+UC groups will be limited to exploratory comparisons.

The American Psychiatric Association guidelines[6] and the Schizophrenia Patient Outcomes Research Team[7] recommend psychosocial interventions in combination with pharmacotherapy. Comprehensive treatment measures include individual and group psychotherapy, psychoeducation, occupational therapy, family therapy and psychosocial interventions such as social skills training.[8]

### Cognitive behavioural therapy for schizophrenia

Cognitive behavioural therapy for psychosis (CBTp) has been shown to reduce positive symptoms and improve coping skills in Europe and the USA.[9] Kuller *et al*[10] reported that CBTp is used in 58% of US medical facilities and 91.3% of UK medical facilities. Wykes *et al*[11] reported an effect size of 0.40 (0.25–0.55) in a meta-analysis of 33 randomised controlled trials (RCTs) (n=1964) on CBTp. Jauhar *et al*[12] found an effect size of 0.33 for general symptoms in a meta-analysis of 34 studies on CBTp for schizophrenia. The National Institute of Health and Clinical Excellence recommends a session structure of 16 sessions or more.[5] However, despite the evidence, the number of practitioners is insufficient; thus, it is challenging for patients to access CBTp.[13] Additionally, CBTp is important because patients' characteristics and problems are diverse due to the illness's nature. It is essential to be flexible in dealing with mood and thoughts, realistic and rational thoughts and delusional and pathological experiences.[14]

There are reports of an association between intrusive thoughts of post-traumatic experiences and schizophrenia.[15 16] There is also a discussion of the potential for CBTp to address intrusive thoughts.[17]

In the future, it will be desirable to establish an effective implementation system for CBTp and increase the number of practitioners in clinical practice.

Evidence of low-intensity CBTp has been collected overseas and its effectiveness has been recognised too.[18 19] Turkington *et al*[20 21] found that community psychiatric nurses' provision of low-intensity CBTp, comprising six sessions, effectively increased awareness of the illness. Overall, six sessions of a 6-week low-intensity CBTp provided by community psychiatric nurses improved the understanding of illness, general status and depression. This suggests that CBTp offered by healthcare professionals may also be effective. Recently, studies have begun investigating CBTp that can be effectively administered online using the internet.[22–25]

### Video-conference cognitive behavioural therapy for psychosis

The number of CBTp practitioners is limited and their use in clinical practice is much less frequent. As a remedy to this problem, there is growing interest in using video-conferenced systems to provide CBT, known as vCBT.[26] This approach has the advantage of providing access to CBT for patients living in remote areas and enabling remote treatment through interactive, real-time communication between the therapist and patient.[27] Although therapist-guided internet-based CBT is shown to be not inferior to face-to-face CBT in treating anxiety, depression, insomnia and somatic disorders,[28] there are less reports about internet-based CBTp. Up to this point in 2022, to the best of our knowledge, no studies have examined the efficacy of individual CBTp using a video-conference system (vCBTp) for schizophrenia.[29]

### Objective

This article describes the study protocol of a RCT designed to evaluate the clinical efficacy of seven sessions of vCBTp, as an adjunct to usual care (UC) compared with UC alone. The trial focuses on patients with schizophrenia who continue experiencing positive symptoms despite receiving pharmacotherapy.

## METHODS AND ANALYSIS

### Study design

The study is designed as a single centre, assessor-blind, two-arm, parallel, prospective, randomised, controlled trial comprising a 7-week treatment regimen. Participants will be allocated to vCBTp plus UC or UC alone (figure 1). We will report according to the Consolidated Standards of Reporting Trials (CONSORT) and the Standard Protocol Items: Recommendations for Interventional Trials reporting guidelines.[30]

### Participants

Inclusion criteria of participants in this study are between 16 and 65 years of age, primary diagnosis of schizophrenia, schizoaffective disorder or delusional disorder according to the fifth edition of the DSM-5, and who are found competent to consent to the study in the MacArthur Competence Assessment Tool for Clinical Research (MacCAT-CR),[31 32] having at least three out of seven points on any one of the seven positive and negative syndrome scale (PANSS) positive symptoms, and those who have taken antipsychotic drugs constantly for 3 months or more and are not planning to change their drugs for the next 3 months. In addition, participants will be required to have internet and computer access and an environment that allows them to take the vCBTp. The exclusion criteria are severe addiction and substance dependence, such as alcohol or drug dependence, intellectual disability, neurocognitive disorders or predicted risk of self-harm or other harm due to worsening symptoms reported by the participants, their family members or their psychiatrists. The risk will be assessed at the time of obtaining consent and at each endpoint by responses to the outcome item and by observation of adverse events. Specifically, patients will be screened for suicidal ideation with nine of the Patient Health Questionnaire-9 (PHQ-9). In particular, those at risk of imminent suicide and are expected to discontinue vCBTp, and those who are hospitalised.

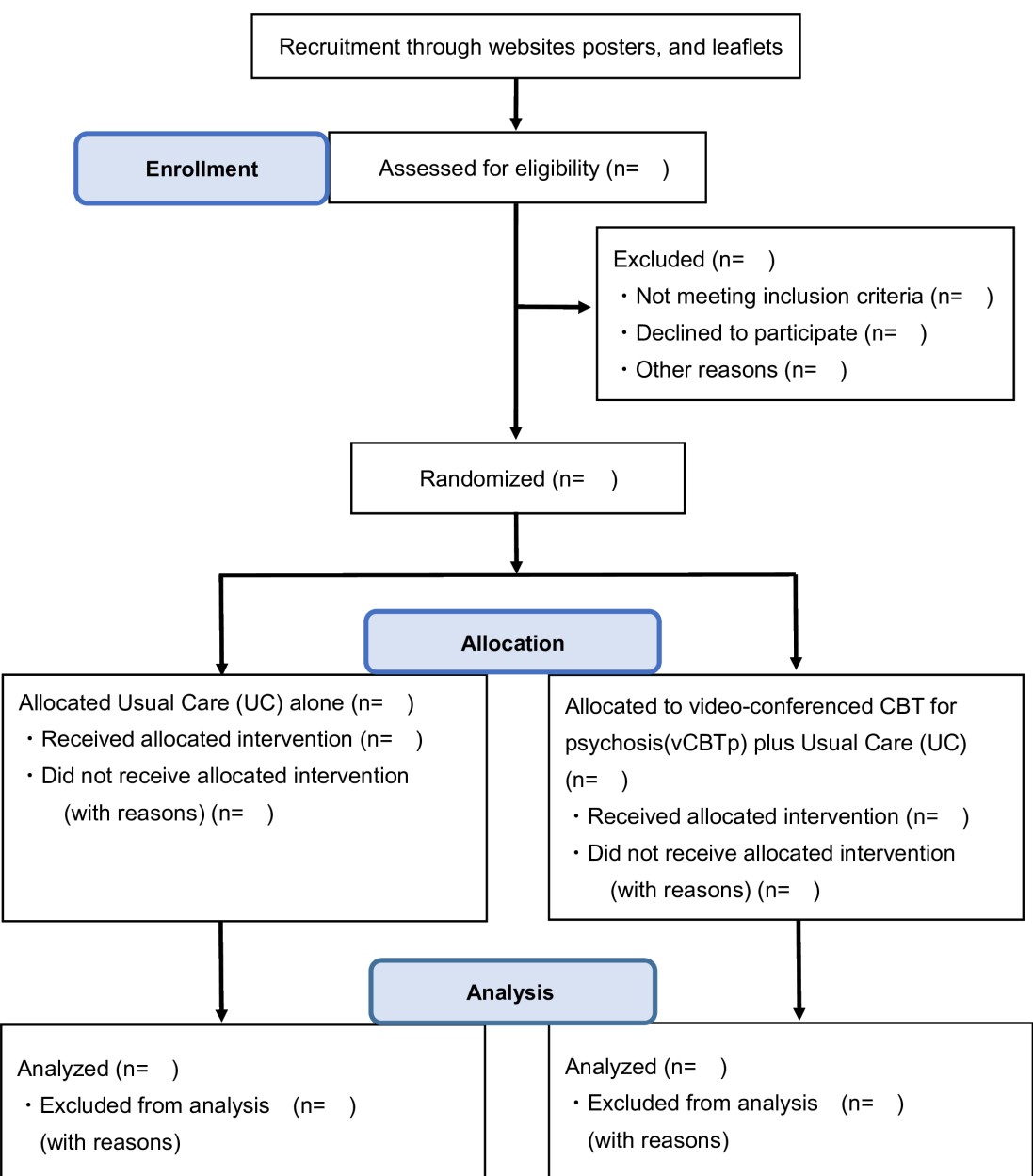

**Figure 1** Flow chart. CONSORT patients' flow diagram of the study, randomisation and treatment. CBT, cognitive behavioural therapy; CONSORT, Consolidated Standards of Reporting Trials.

Two researchers (a psychiatrist (ES) and a researcher (MK)) will evaluate and confirm the patients' eligibility including diagnosis, treatment history by their psychiatrists, the suitability of the symptoms and their ability to consent to participate in the study.

### Details of recruitment

The researchers will recruit participants through websites, posters and leaflets placed at medical institutions in Chiba prefecture, Japan, from April 2021 to March 2025 until 24 participants will be enrolled. Participants will be required to obtain permission from their psychiatrists before enrolment in the study and continue to receive medical care including pharmacotherapy from their psychiatrists. The study will be conducted in the Cognitive Behavioural Therapy Center outpatient clinic at Chiba University Hospital.

### Intervention methods

Participants will not be restricted to changes in their medications by their psychiatrists during the study period. If clinically appropriate, they will be allowed to participate in usual psychosocial intervention programmes such as social skills training at psychiatric daycare or night care.

However, to properly evaluate the effectiveness of vCBTp, participants may not receive other CBT or specific psychotherapies including psychoanalytical therapy, Morita therapy, insight therapy, hypnotherapy, brain stimulation therapies including electroconvulsive therapy or magnetic stimulation therapies. The vCBTp+UC group

will receive the text material developed by the research team by mail. All seven sessions will be conducted according to the text material. The therapist will assign homework to the participants at the end of each session, as described in the text material.

### Video-conference cognitive behavioural therapy programme for schizophrenia

The vCBTp programme for schizophrenia was developed by two researchers (ES and MK). The number of sessions was set at 7, as the median number of sessions was 7.5 in a meta-analysis of low-intensity CBTp.[18] The treatment lasts over 7 weeks (once a week, 50 min per session). It included elements incorporated into CBT for depression and anxiety, in general. This protocol primarily focuses on cognitive restructuring. The components are as follows: (1) Assessment and goal setting, (2) externalisation of stress in present life, (3) cognitive restructuring of stress in present life, (4) mood change and relaxation, (5) externalisation of past stressful experiences, (6) reappraisal of past stressful experiences and (7) relapse prevention. As described in the introduction, we focus on subjective distress from intrusive thoughts of stressful experiences in patients with schizophrenia.[15–17] We allow flexibility on changing or repeating the order of sessions depending on participants' understanding and reactions.

Each session includes examples of cognitive restructuring for patients with schizophrenia and homework assignments. The therapist conducting the session and the participant will be connected remotely in real-time via a web conferencing system between the participant's house and Chiba University Hospital for CBT. Participants will have to work on the vCBTp and receive supportive feedback on the requested homework. Participants will also be allowed to email the research office (MK) if they have questions regarding the contents.

### Outcomes

#### Baseline and clinical characteristics

The baseline and clinical characteristics included sex, age, marital status, employment status, age at onset of schizophrenia, duration of illness and treatment history including the name of the antipsychotic to which the patient has developed resistance, current drug titres at baseline, and any changes in conventional treatment during the study period will be collected by researchers.

#### Primary outcome

The primary outcome is change in the PANSS[33] total score at week 8 from baseline. The PANSS is a 30-item interview-based assessment consisting of 7 positive symptoms, 7 negatives and 16 general psychopathology items. Each subscale is scored on a scale of 1–7 points. The PANSS is an objective scale for assessing schizophrenia symptoms in clinical and experimental studies and has become a worldwide standard for reliability and validity. An assessor (a researcher, HN) blind to treatment allocation will evaluate the PANSS by web conferencing system

at baseline and at week 8, respectively. Comparative results between video-conference and face-to-face assessments for schizophrenia are similar.[34] Furthermore, it has been suggested that there is no difference between face-to-face and remote assessments on the PANSS.[35]

#### Secondary outcomes

The secondary outcomes are the following:

We will evaluate the sum of positive symptoms, negative symptoms and general psychopathology extracted from the PANSS subscales score, respectively. The total score of the seven positive symptoms subscale on the PANSS ranges from 7 (no symptoms) to 49 (severe positive symptoms). The total score of the seven negative symptoms subscale ranges from 7 (no symptoms) to 49 (severe negative symptoms). The total score of the 16 items of general psychopathology subscale ranged from 16 (no symptoms) to 112 (severity of psychopathology).

The Japanese version of the Beck Cognitive Insight Scale (BCIS-J)[36] will be used to assess cognitive insight. A total of 15 items of the BCIS-J are divided into 9 items of self-reflection (0–36) and 6 items of self-confidence (0–24). The lower the subtotal of 9 self-reflection items minus 6 self-confidence items, the lower the cognitive pathology.

The Impact of Event Scale-Revised (IES-R)[37] will be used to assess subjective distress for a specific stressful life event. The 22 items (0–88) of IES-R comprise eight intrusive symptoms, eight avoidance symptoms and six hyperarousal symptoms, and can measure symptoms in persons exposed to traumatic experiences.

The PHQ-9[38] will be used to assess depressive symptoms. PHQ-9 comprises nine items ranging from 0 (no depressive symptoms) to 27 (severe depressive symptoms).

The Generalised Anxiety Disorder-7 (GAD-7)[39] will be used to assess anxiety. GAD-7 comprises seven items ranging from 0 (no anxiety symptoms) to 21 (severe anxiety symptoms).

The 5-level EuroQol 5-dimensional questionnaire[40] will be used to assess quality of life. It comprises five items and assesses the quality of life on a 5-point Likert scale from 1 (not severe) to 5 (severe). It is the most commonly used measure of quality-adjusted life-years worldwide.

In addition, we will evaluate the chlorpromazine equivalent to daily prescriptions for antipsychotic medications.[41]

Therapists will ask the participants about their experience of adverse events during each assessment. All measures will be assessed at week 0 (baseline) and week 8 (post-intervention), and the results will be analysed per the intention-to-treat principle.

### Sample size

Primarily, this study aims to test the superiority of vCBTp on symptom improvement in a group of patients with schizophrenia compared with a UC group. We refer to a previous study by Morrison et al[42] in which CBT and antipsychotic groups were compared. Assuming that the mean difference between the vCBTp plus UC group and

UC alone group after the intervention is 12 points and the SD of the pooled UC group is 8, the number of patients in each group was calculated as 11 for a significance level of 5% on both sides and a power of 90%. Therefore, the target number of patients was set at 12 in each group, assuming a dropout rate of 10% for 24 patients. Furthermore, a previous study suggested that 12 patients in each group is an appropriate number in a pilot study.[43]

### Randomisation and assessor-blindness
After baseline assessment, participants will be randomly assigned to the UC group or vCBTp+UC in a 1:1 ratio, using the minimisation method to ensure a balance between baseline PANSS total score (PANSS ≥51) and sex. The PANSS values are based on a study by Naeem et al[44] Each participant will be randomly assigned to one of the two treatments. The assessor (a researcher, HN) will be not informed of the participant's allocation group throughout the study period for assessor-blindness.

### Data analysis plan
The study will follow the CONSORT guidelines for statistical analysis and reporting. The primary analysis will be based on the intention-to-treat principle. To evaluate the primary outcome, the changes in the PANSS total score at week 8 will be calculated for both the UC group and the vCBTp+UC group. A comparison between the groups will be performed using an unpaired t-test. Secondary outcomes will also be analysed to provide further understanding of the primary endpoints, following a similar approach as the primary outcome analysis. In addition, the number of subjects in each group who have improved by 20% of the total PANSS score are considered to be treatment-responsive and the proportion of subjects in each group is calculated. The $\chi^2$ test is used to compare proportions between groups. The safety endpoint is the frequency of adverse events, and a tabulation table is prepared for the endpoints with exact two-sided 95% CIs of the binomial distribution calculated for each group for percentage estimates. If necessary, a comparison between groups is made using the Fisher's direct probability calculation method. The statistical analyses will be performed using SAS V.9.4 (SAS Institute, Cary, North Carolina, USA), and a p value of <0.05 will be considered statistically significant.

### Ethics and dissemination
The study protocol has been approved by the Institutional Review Board of the Chiba University Hospital (reference number: G2020031) in January 2021.

We will conduct this study at the outpatient department of the Cognitive Behavioural Therapy Center at Chiba University Hospital. Prospective participants will be informed of the purpose of the study and asked about their willingness to participate when they contact the researchers. Participation is voluntary and complete anonymity will be guaranteed. Further, participants would be notified that they could drop out of the study at any point. Each participant will be asked to provide written informed consent to participate in the study after MacCAT-CR confirms that the participant is competent to consent to the clinical trial. All participants will receive UC from their psychiatrists, and half of the participants will receive vCBTp in addition to UC. Participants randomised to the UC arm will be eligible to receive vCBTp in an ancillary study (UMIN000044244) after completion of the study. A blinded assessor will evaluate the participants at each assessment point (Weeks 0 and 8). All adverse events will be reported and serious ones will be immediately reported to the Clinical Trial Review Committee of Chiba University Hospital. Moreover, they will be registered in the hospital's risk management system. An independent data monitoring committee then properly reviews detailed records of the progress of the clinical trial, key efficacy variables and safety data and recommends the trial's continuation, modification or termination to the clinical investigators accordingly. Regardless of the outcome, the trial results will be published in an international journal. This study will be conducted and reported per CONSORT recommendations.

### Patient and public involvement
Patients and the public were not involved in the design or conduct of the study.

## DISCUSSION
This study will address the effect of vCBTp for patients with schizophrenia who continue experiencing positive symptoms despite receiving pharmacotherapy.[13 20 21 45 46] The findings of this study will provide valuable evidence to facilitate the development of new therapeutic support modalities and promote increased treatment options for patients. The anticipated limitations of this study are as follows. First, because this study is a single-centre study, it has less generalisability than those conducted in multicentre studies. Second, the follow-up study is set up as a separate study. Therefore, verifying the retention of this effect in this study is not easy. The present study is a pilot study and the study design is based on consideration of patient burden. If this study is successfully completed without any serious adverse events resulting in hospitalisation or potentially leading to disability, and if a certain level of efficacy is suggested, it would be advisable to plan a full-scale study with a larger sample size.

### Author affiliations
[1]Departments of Cognitive Behavioral Physiology, Chiba University Graduate School of Medicine School of Medicine, Chiba, Japan
[2]Department of Rehabilitation, Faculty of Health Care and Medical Sports, Teikyo Heisei University - Chiba Campus, Ichihara, Japan
[3]Department of Nursing, Faculty of Medicine, Jikei University School of Medicine, Minato-ku, Japan
[4]Clinical Research Center, Chiba University Hospital, Chiba, Japan
[5]Future Medicine Research Center, Chiba University, Chiba, Japan
[6]Department of Psychiatry and Psychosomatic Medicine, Kameda Medical Center, Kamogawa, Japan

# Open access

[7]Department of Cognitive Behavioral Physiology, Graduate School of Medicine, Chiba University, Chiba, Japan
[8]Research Center for Child Mental Development, Graduate School of Medicine, Chiba University, Chiba, Japan

**Acknowledgements** We thank all the staff at the Cognitive Behavioral Therapy Center of Chiba University Hospital. We are grateful to all participants in this study. We would like to thank Editage (www.editage.com) for English language editing.

**Contributors** MK and ES designed the study and developed the protocol with HH and YS. MK, ES and HK recruited the patients. MK, HN and ES managed the preparation of the manuscript. MK wrote the first draft of the manuscript. MK, HN and ES wrote the manuscript. All the authors contributed to and approved the final version of the manuscript.

**Funding** This work was supported by a Grant-in-Aid for Scientific Research (Grant No. 21K11197, Japan Society for the Promotion of Science).

**Competing interests** None declared.

**Patient and public involvement** Patients and/or the public were not involved in the design, or conduct, or reporting, or dissemination plans of this research.

**Patient consent for publication** Not applicable.

**Provenance and peer review** Not commissioned; externally peer reviewed.

**ORCID iDs**
Masayuki Katsushima http://orcid.org/0000-0002-3605-5156
Yuki Shiko http://orcid.org/0000-0002-3959-9343
Eiji Shimizu http://orcid.org/0000-0002-6741-9338

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
