## [Reviewer comments · BMJ Open]

ARTICLE DETAILS

TITLE (PROVISIONAL)	Randomised controlled trial on the effect of video-conference cognitive behavioural therapy for patients with schizophrenia: A study protocol
AUTHORS	Katsushima, Masayuki; Nakamura, Hideki; Hanaoka, Hideki; Shiko, Yuki; Komatsu, Hideki; Shimizu, Eiji

VERSION 1 – REVIEW

REVIEWER	Sumiyoshi, Tomiki National Center of Neurology and Psychiatry
REVIEW RETURNED	18-Feb-2023

GENERAL COMMENTS	This is a study protocol to compare the effect of pCBT in a face-to-face fashion and that through video- conference. The prospective data obtained through the proposed study would provide an insight into the promotion of pCBT. There is a contradictory discussion as to the sample size. In Abstract, they raise concern the small sample size that is planned in this protocol. In Discussion, they again raise the limitation that would be produced by the inappropriately small sample size to be used. Why, then, did they make the present protocol? I would suggest the change of the sample size, or substantially change the aim of this study.
---

REVIEWER	Ishigaki, Takuma Tokyo University of Science - Kagurazakakudan Campus
REVIEW RETURNED	03-Apr-2023

GENERAL COMMENTS	I believe this study is an important clinical study in the treatment of schizophrenia. I would like to ask you to answer a few questions about its assessment and intervention methods. 1. Please write a rationale why the PANSS online assessment is of the same quality as the face-to-face assessment.2. This treatment consists of 7 sessions, is there any reason for the authors to believe that 7 sessions are sufficient?3. Since CBTp often requires flexibility in sessions, does this treatment change or repeat the order of sessions depending on participants' understanding and responses? Or does it not change the set order?
--

REVIEWER	Mason, Ava University College London, Division of Psychiatry
REVIEW RETURNED	19-Apr-2023

GENERAL COMMENTS	Abstract
----------

	Within the first line of the introduction section, more detail is required to describe what CBTp has proven effective in doing eg. Reducing psychotic symptom severity, improving functioning. In the methods section, 'UC' should be written as usual care before the abbreviation is used. Introduction The introduction was well structured, with a clear rationale behind the aims of the protocol. Within the standard treatment for schizophrenia section, a little more detail could have been provided on what type of pharmacotherapy is offered as a first treatment option for those with schizophrenia. Methods and analysis Within the participant's section, more detail should have been provided on what would objectively be considered a high risk of self-harm by participants, family members, or psychiatrists. Also, it was unclear what would be classified as resistance to pharmacotherapy within this section or when mentioning resistance in the first outcome paragraph. The recruitment and intervention methods sections were clearly explained. There are spelling mistakes in 'introduction' on page 4 (or 9/28) line 15 and 'allocation' on page 4 line 51. Data analysis plan The data analysis plan should have explained how secondary outcomes would be analysed in more detail in order for replication to be possible. Page 7 line 7 change of font halfway through the sentence. Ethics and dissemination In line 48 when stating participation is voluntary, emphasis could have been provided to say that participants would be notified that they can drop out of the study at any point. Discussion This discussion section generally described the relevant strengths and limitations of the study protocol. However, it was unclear how the safety of the study would be assessed in order to consider whether a follow-up study would be conducted.
--	--

VERSION 1 – AUTHOR RESPONSE

Reviewer: 1

Dr. Tomiki Sumiyoshi, National Center of Neurology and Psychiatry

Comments to the Author:

This is a study protocol to compare the effect of CBTp in a face-to-face fashion and that through video- conference. The prospective data obtained through the proposed study would provide an insight into the promotion of CBTp.

Response

Thank you for the review and all your input. We will further test the effectiveness of CBTp for dissemination in the future.

There is a contradictory discussion as to the sample size. In Abstract, they raise concern the small sample size that is planned in this protocol. In Discussion, they again raise the limitation that would be produced by the inappropriately small sample size to be used. Why, then, did they make the present protocol? I would suggest the change of the sample size, or substantially change the aim of this study.

Response 4

The sample size in this study protocol was estimated and estimated by a biostatistician. The appropriate number of cases in the pilot study published in a previous study (Julious, 2005) was also used as a reference. Therefore, we have revised the description of the small sample size in the abstract and discussion.

Furthermore, a previous study suggested that 12 patients in each group is an appropriate number in a pilot study.⁴³

(43. Julious SA. Sample size of 12 per group rule of thumb for a pilot study. *Pharmaceut Statist* 2005;4:287-291.)

if a certain level of efficacy is suggested, it would be advisable to plan a full-scale study with a larger sample size.

Reviewer: 2

Prof. Takuma Ishigaki, Tokyo University of Science - Kagurazakakudan Campus

Comments to the Author:

I believe this study is an important clinical study in the treatment of schizophrenia. I would like to ask you to answer a few questions about its assessment and intervention methods.

Response

Thank you for your peer review and suggestions. I would like to continue my research in order to contribute to patients with schizophrenia patients.

1. Please write a rationale why the PANSS online assessment is of the same quality as the face-to-face assessment.

Response 5

Since previous studies have shown that face-to-face and remote assessments using videoconferencing systems are equivalent for patient with schizophrenia and that the PANSS is not dependent on face-to-face assessments and can also be assessed on a recording basis, we determined that remote PANSS assessments in this study were also feasible.

Comparative results between video-conference and face-to-face assessments for schizophrenia are similar.³⁴ Furthermore, it has been suggested that there is no difference between face-to-face and remote assessments on the PANSS.³⁵

(34. Sharp IR, Kobak KA, Osman DA. The use of videoconferencing with patients with psychosis: a review of the literature. *Ann Gen Psychiatry* 2011;10:14.)

(35. Targum SD, Pendergrass JC, Murphy C. Audio-digital recordings to assess ratings reliability in clinical trials of schizophrenia. *Schizophr Res* 2021;232:54-60.)

2. This treatment consists of 7 sessions, is there any reason for the authors to believe that 7 sessions are sufficient?

Response 6

According to the literature (Hazell et al., 2016), 10 articles were cited in the meta-analysis on low-intensity CBTp. Since the median number of sessions for the 10 studies was 7.5 sessions, we set the session structure for this study to 7 sessions.

The number of sessions was set at seven, as the median number of sessions was 7.5 in a meta-analysis of low-intensity CBTp.¹⁸

(18. Hazell CM, Hayward M, Cavanagh K, et al. A systematic review and meta-analysis of low intensity CBT for psychosis. *Clin Psychol Rev* 2016;45:183-192.)

3. Since CBTp often requires flexibility in sessions, does this treatment change or repeat the order of sessions depending on participants' understanding and responses? Or does it not change the set order?

Response 7

The seven-session structure was used as the basis for the session progression, but the order could be changed or repeated in some cases, depending on the participants' understanding and reactions.

We allowed flexibility on changing or repeating the order of sessions depending on participants' understanding and reactions.

Reviewer: 3

Ms. Ava Mason, University College London, Kings College London

Comments to the Author:

Abstract

Within the first line of the introduction section, more detail is required to describe what CBTp has proven effective in doing e.g. Reducing psychotic symptom severity, improving functioning.

Response 8

Thank you for your peer review and suggestions. The first line of the Introduction has been revised as follows.

Cognitive behavioural therapy for psychosis (CBTp) has demonstrated effectiveness in reducing

positive symptoms, improving depression, enhancing coping skills, and increasing awareness of illness.

In the methods section, 'UC' should be written as usual care before the abbreviation is used.

Response 9

We have revised our description of "usual care" to read.

Introduction

The introduction was well structured, with a clear rationale behind the aims of the protocol. Within the standard treatment for schizophrenia section, a little more detail could have been provided on what type of pharmacotherapy is offered as a first treatment option for those with schizophrenia.

Response 10

We have added the following statement regarding initial pharmacotherapy for schizophrenia.

Antipsychotic medications are commonly used for patients with schizophrenia primarily to reduce positive symptoms, such as hallucinations and delusions by regulating the activity of dopamine and other nerves in the brain. In recent years, atypical antipsychotic prescriptions have been promoted. They effectively improve positive and negative symptoms and cognitive function but may cause side effects in some patients.

Methods and analysis

Within the participant's section, more detail should have been provided on what would objectively be considered a high risk of self-harm by participants, family members, or psychiatrists.

Response 11

We added the following sentence as a description of objectively assessing the presence or absence of risk for self-harm.

The risk will be assessed at the time of obtaining consent and at each endpoint by responses to the outcome item and by observation of adverse events. Specifically, patients will be screened for suicidal ideation with nine of the PHQ-9.

Also, it was unclear what would be classified as resistance to pharmacotherapy within this section or when mentioning resistance in the first outcome paragraph.

Response 12

We have revised the description of the study to read "schizophrenic patients whose symptoms do not remit after pharmacotherapy" as follows.

The trial focuses on patients with schizophrenia who continue experiencing positive symptoms despite receiving pharmacotherapy.

The recruitment and intervention methods sections were clearly explained.

There are spelling mistakes in 'introduction' on page 4 (or 9/28) line 15 and 'allocation' on page 4 line 51.

Response 13

Thanks for pointing this out. We have corrected the spelling error.

introduction

allocation

Data analysis plan

The data analysis plan should have explained how secondary outcomes would be analysed in more detail in order for replication to be possible.

Response 14

The following amendments were made to the data analysis for the primary and secondary endpoints to explain in more detail how the data would be analysed.

The study will follow the CONSORT guidelines for statistical analysis and reporting. The primary analysis will be based on the intention-to-treat principle. To evaluate the primary outcome, the changes in the PANSS total score at week 8 will be calculated for both the UC group and the vCBTp+UC group. Using an unpaired t-test, a comparison between the groups will be performed. Following a similar approach as the primary outcome analysis, secondary outcomes will also be analyzed to provide a further understanding of the primary endpoints. Furthermore, the number of subjects in each group who have improved by 20% of the total PANSS score is considered to be treatment-responsive, and the proportion of subjects in each group is calculated. The χ^2 test is used to compare proportions between groups. The safety endpoint is the frequency of adverse events, and a tabulation table is prepared for the endpoints, with exact two-sided 95% confidence intervals of the binomial distribution calculated for each group for percentage estimates. A comparison between groups is made using Fisher's direct probability calculation method, if necessary. The statistical analyses will be performed using SAS ver. 9.4 (SAS Institute Inc., Cary, NC, USA), and a p-value of less than 0.05 will be considered statistically significant.

Page 7-line 7 change of font halfway through the sentence.

Response

Thanks for pointing this out. The font has been corrected Times New Roman.

Ethics and dissemination

In line 48 when stating participation is voluntary, emphasis could have been provided to say that participants would be notified that they can drop out of the study at any point.

Response 15

We added that participants will be notified that they can drop out of the study at any time.

Further, participants would be notified that they could drop out of the study at any point.

Discussion

This discussion section generally described the relevant strengths and limitations of the study protocol. However, it was unclear how the safety of the study would be assessed in order to consider whether a follow-up study would be conducted.

Response 16

Safety is ascertained by the presence or absence of adverse events during participation. If there are no serious adverse events, safety is considered ensured. The absence of serious adverse events will be considered "safe".

If this study is successfully completed without any serious adverse events resulting in hospitalization or potentially leading to disability,

VERSION 2 – REVIEW

REVIEWER	Sumiyoshi, Tomiki National Center of Neurology and Psychiatry
REVIEW RETURNED	08-Aug-2023

GENERAL COMMENTS	I think the paper has been adequately revised.
--

REVIEWER	Ishigaki, Takuma Tokyo University of Science - Kagurazakakudan Campus
REVIEW RETURNED	25-Jul-2023

GENERAL COMMENTS	The reviewer determined that the paper has been appropriately revised in accordance with my requirements.
---